# Assessment of the Risk of Oral Cancer Incidence in A High-Risk Population and Establishment of A Predictive Model for Oral Cancer Incidence Using A Population-Based Cohort in Taiwan

**DOI:** 10.3390/ijerph17020665

**Published:** 2020-01-20

**Authors:** Li-Chen Hung, Pei-Tseng Kung, Chi-Hsuan Lung, Ming-Hsui Tsai, Shih-An Liu, Li-Ting Chiu, Kuang-Hua Huang, Wen-Chen Tsai

**Affiliations:** 1Department of Public Health, China Medical University, Taichung 40402, Taiwan; olivehung@gmail.com; 2Department of Healthcare Management, Yuanpei University of Medical Technology, Hsinchu 30015, Taiwan; 3Department of Healthcare Administration, Asia University, Taichung 41354, Taiwan; ptkung@asia.edu.tw; 4Department of Medical Research, China Medical University Hospital, China Medical University, Taichung 40402, Taiwan; 5Department of Social Work, National Quemoy University, Quemoy 892, Taiwan; hsuan@nutc.edu.tw; 6Department of Otolaryngology, China Medical University Hospital, Taichung 40447, Taiwan; minghsui@mail.cmuh.org.tw; 7Department of ENT, Taichung Veterans General Hospital, Taichung 40705, Taiwan; saliu@vghtc.gov.tw; 8Department of Health Services Administration, China Medical University, Taichung 40402, Taiwan; litingchiu933@gmail.com (L.-T.C.); khhuang@mail.cmu.edu.tw (K.-H.H.)

**Keywords:** oral cancer, smoking, betel nut chewing, screening, prediction

## Abstract

We aimed to assess the risk of oral cancer incidence in a high-risk population, establish a predictive model for oral cancer among these high-risk individuals, and assess the predictive ability of the constructed model. Individuals aged ≥30 years who had a habit of smoking or betel nut chewing and had undergone oral cancer screening in 2010 or 2011 were selected as study subjects. The incidence of oral cancer among the subjects at the end of 2014 was determined. The annual oral cancer incidence among individuals with a positive screening result was 624 per 100,000 persons, which was 6.5 times that of the annual oral cancer incidence among all individuals screened. Male sex, aged 45–64 years, divorce, low educational level, presence of diabetes, presence of other cancers, high comorbidity severity, a habit of smoking or betel nut chewing, and low monthly salary were high-risk factors for oral cancer incidence (*p* < 0.05). The area under the curve of the predictive model for oral cancer incidence was 0.73, which indicated a good predictive ability. Therefore, the oral cancer screening policy for the high-risk population with a habit of smoking and/or betel nut chewing is beneficial for the early diagnosis of oral cancer.

## 1. Introduction

There are about 657,000 new cases of oral and throat cancer worldwide each year, resulting in more than 330,000 deaths; based on the exposure to risk factors, oral cancer in Central and South Asia is relatively severe [1]. According to the International Agency for Research on Cancer, some Asia-Pacific countries had the top three rates of oral cancer in 2018 [2,3,4,5]. The crude incidence rate of oral cancer in Taiwan is 32.46 per 100,000 persons, which is the highest worldwide [6]. Oral cancer is also the most common type of head and neck cancer in Taiwan, with the incidence rate among men being 10.9 times higher than that among women [6]. By comparing the incidence rates of various cancers in Taiwan, it can be observed that oral cancer ranks fifth among the top 10 causes of deaths due to cancer and has been the fourth most common cancer among men for 12 consecutive years since 2003 [6]. Therefore, the prevention and control of oral cancer are regarded as key health issues in Taiwan. 

The betel nut chewing culture unique to Asians is also a key risk factor that has been highlighted by many researchers [7,8,9,10,11,12,13,14]. In a report by the International Agency for Research on Cancer (IARC), betel nut has been cited as a group 1 carcinogen [15]. As approximately 86% of Taiwanese oral cancer patients are habitual betel nut chewers and betel nuts are readily available in Taiwan [6], a greater amount of attention should be focused on the health hazards associated with betel nut chewing.

According to statistics from the 2015 Cancer Registry Annual Report issued by the Health Promotion Administration of the Ministry of Health and Welfare of Taiwan, the risk of cancer incidence is higher among Taiwanese men than among Taiwanese women, and the age-specific cancer incidence increases gradually with age [6]. Similarly, the global age-standardized incidence rate (ASIR) for all cancers per 100,000 persons among men is significantly higher than that among women, and the cancer incidence rates for both men and women increase with age [16]. 

Health-related behaviors are closely linked to oral cancer, with excessive drinking, smoking, and betel nut chewing leading to an increased risk of oral cancer incidence [17,18,19,20,21,22,23], and HPV infection, poor oral health, Lichen Planus, etc. have also been pointed out in many literatures as risk factors for oral cancer [22,24,25,26,27,28,29]. In India, chewing tobacco is the strongest predictor of upper gastrointestinal cancer, and its effect is even greater than those of drinking or smoking. The combined effects of smoking and drinking increases this population’s risk of cancer by a factor of 12 when compared to those who have never smoked or drank [30]. Data from pioneering research by Taiwanese researchers indicate that the risk of oral cancer incidence among individuals with concurrent habits of smoking, drinking, and betel nut chewing is 123 times higher than that among the general population. However, there was only a statistically significant association between oral cancer and chewing betel nuts [13,31]. In many studies, researchers have asserted that socioeconomic status (SES), which includes educational level, income, and occupation, is also linked to the risk of oral cancer incidence [32,33,34,35,36]. Generally, the risk of oral cancer incidence is lower in individuals with higher educational and income levels. However, with the progression of time, the difference in the risk of oral cancer incidence between high-income earners and low-income earners has gradually reduced [36]. Regarding the degree of urbanization, statistics reported by the American Cancer Society have indicated that the ASIR for all cancers per 100,000 persons in highly developed regions is higher than the corresponding cancer incidence rate in less developed regions for both men and women [37], which shows that regions with a higher level of urbanization have a greater risk of cancer incidence. In North America and Europe, “high-risk” human papillomavirus infection is responsible for the increasing percentage of young people with oropharyngeal cancer [22,26]. Relevant research has indicated that HPV has a statistically significant correlation with oral cancer [22,24,26,28,29]. The prevalence of HPV in oropharyngeal head and neck cancer (OPC) has increased over time, with an increase within 10 years from 40.5% before 2000 to 72.2% in 2005–2009 [26,27], showing that HPV infection is an important indicator of oral cancer risk. In addition, relationships between poor oral hygiene, periodontal disease, chronic candidiasis, herpesvirus infection, and oral cancer have been proposed in many studies [25,27].

In the present study, by analyzing national data on oral cancer screening in Taiwan with consideration of the time factor, we attempted to assess the risk of oral cancer incidence in a high-risk population, establish a predictive model for the risk of oral cancer incidence among these individuals, and assess the predictive ability of the established model. 

## 2. Materials and Methods 

### 2.1. Study Subjects

Individuals aged 30 years and above who had a habit of smoking or betel nut chewing and had underwent oral cancer screening in 2010 or 2011 were followed up to the end of 2014 to determine the oral cancer morbidity status. All subjects included in the oral cancer screening files for 2010 and 2011 were included in the analysis. Only those who had oral cancer before the screening were excluded.

### 2.2. Data Sources

The data used in the present study were obtained from databases and records provided by the Department of Statistics of the Ministry of Health and Welfare of Taiwan, which included oral cancer screening records [38], cancer registry records [39], the National Health Insurance Database, and household registration records from the Ministry of the Interior of Taiwan. The criteria for extracting variables from the database are based on the demographic variables and health-related factors related to the incidence of oral cancer shown in previous literatures [11,27,33,40,41,42,43,44,45,46,47,48,49,50,51,52,53,54,55,56,57]. First, the basic data, oral screening data, and data on health-related behaviors of the subjects were extracted from the cancer screening records. This was followed by the extraction of data such as date and method of confirmed diagnosis of oral cancer from the cancer registry records and the extraction of data on other catastrophic illnesses from the Registry for Catastrophic Illness Patients database. Data on variables such as sex, age, SES, health status and utilization of preventive health services were extracted from the National Health Insurance Database. Other data such as those regarding the degree of urbanization, educational level, marital status, and race were extracted from the household registration records [58].

### 2.3. Definition of Variables

The variables used in the present study included the following: sex, age, educational level, marital status, indigenous race, degree of urbanization of the place of residence, presence or absence of diabetes, presence or absence of other cancers [11,27,33,40,41,42,43,44,45,46,47,48,49,50,51,52,53,54,55,56,57], Charlson comorbidity index (CCI), presence or absence of other catastrophic illnesses, health-related behaviors, and monthly salary. The degrees of urbanization of the places of residence of the participants were classified into seven levels, with level 1 corresponding to regions with the highest degree of urbanization and level 7 corresponding to regions with the lowest degree of urbanization. The CCI proposed by Deyo et al. was used as a measure of comorbidity severity [59]. The presence or absence of catastrophic illnesses was determined based on the definition of catastrophic illnesses stated by the National Health Insurance Administration of Taiwan in the Registry for Catastrophic Illness Patients, which includes 30 categories of diseases such as cancer, stroke, and renal disease requiring dialysis [60]. Individuals with other catastrophic diseases other than malignant neoplasms were classified as having other catastrophic illnesses [60]. The health-related behavior categories included smoking, betel nut chewing, and concurrent smoking and betel nut chewing habits.

### 2.4. Statistical Analysis

A univariate Poisson regression analysis was first performed and the incidence density ratio (IDR) was used to indicate the annual risk of oral cancer incidence per 100,000 persons. Subsequently, the log-rank test was used to analyze the correlations between the various factors and risk of oral cancer incidence, and a Cox proportional hazards (PH) model was used to examine the risk factors and curve for the risk of oral cancer incidence. The onset of oral cancer before the end of 2014 was defined as an event and the observation period was calculated up to the time of onset. The non-occurrence of oral cancer before the end of 2014 was defined as a censor and the observation period was assessed up to the end of 2014.

We used a logistic regression model as a predictive model, and the area under the receiver operating characteristic (ROC) curve (AUC) was determined to assess the ability of the model to predict the risk of oral cancer incidence. The ROC curve was generated by plotting sensitivity (i.e., true positive rate) against 1—specificity (i.e., false positive rate). The AUC is a measure of the classification ability of the model. AUC values are generally interpreted as follows, with higher values indicating better classification ability: <0.5: test not useful, 0.5–0.6: bad, 0.6–0.7: sufficient, 0.7–0.8: good, 0.8–0.9: very good, 0.9–1.0: excellent [61].

## 3. Results

In the present study, the annual cancer morbidity per 100,000 persons among the Taiwanese population with smoking and betel nut chewing habits who had undergone oral cancer screening was measured using the IDR. The annual oral cancer incidence rate among the screened individuals was 96 per 100,000 persons (Table 1). The incidence rate among men was 5.5 times higher than that among women, and the incidence rate among individuals aged 45–64 years was significantly higher than that among individuals aged <45 years. Notably, close to 5% of oral cancer patients were aged below 45 years; this is indicative of a rising cancer incidence among the younger population, which warrants attention. When the relationships between incidence rate and the various SES-related factors were examined, we found that the incidence rate decreased with increasing educational level, divorcees had a significantly higher incidence rate than married individuals, individuals residing in regions with a lower degree of urbanization had a higher incidence rate, and the incidence rate decreased with increasing income level. Among individuals with different health-related behaviors, the risk of oral cancer incidence among betel nut chewers was significantly higher than that among smokers. Individuals who had concurrent habits of smoking and betel nut chewing had a higher oral cancer risk than individuals who only had either a smoking habit or a betel nut chewing habit. The incidence of oral cancer among individuals who were diabetic prior to screening was 1.62 times higher than that among those who were not diabetic. Individuals with other cancers prior to screening had an oral cancer incidence rate that was 2.13 times higher than that among individuals without other cancers. The cancer incidence rate increased with comorbidity severity and was higher among individuals with other catastrophic illnesses than among those without other catastrophic illnesses.

When the data of individuals with positive screening results were compared with those of all screened individuals, we found that the annual oral cancer incidence rate among individuals with positive screening results was 624 per 100,000 persons, which was 6.5 times higher than that among all screened individuals (96 per 100,000 persons). This indicates that screening is indeed beneficial toward identifying the population with a high-risk for oral cancer incidence. If timely medical consultations and continuous follow-up can be provided to the population with positive screening results, the risk of oral cancer incidence in these individuals should be effectively reduced.

By considering the time of cancer incidence, the relationships between the various variables and the risk of cancer incidence were determined using the log-rank test. Sex, age, educational level, marital status, degree of urbanization of the place of residence, presence or absence of diabetes, presence or absence of other cancers, comorbidity severity, presence or absence of other catastrophic illnesses, smoking and betel nut chewing habits, and monthly salary were correlated with the risk of oral cancer incidence in screened individuals (*p* < 0.05) (Table 2).

Using a Cox PH model, we investigated the risks of oral cancer incidence in individuals screened for oral cancer and the relevant factors. Sex, age, educational level, marital status, presence or absence of diabetes, presence or absence of other cancers, comorbidity severity, smoking and betel nut chewing habits, and monthly salary influenced oral cancer incidence among individuals who had undergone oral cancer screening (Table 3). Among the individuals screened, men had a higher risk of oral cancer incidence (hazard ratio (HR) = 5.72). In particular, the risk of oral cancer incidence in individuals with an educational level of junior high school and above was 0.37–0.89 times higher than that among individuals with an educational level of elementary school and below. Individuals with a habit of betel nut chewing (HR = 2.12) or concurrent habits of smoking and betel nut chewing (HR = 2.29) also had a higher risk of oral cancer incidence than individuals who only had the habit of smoking. 

Figure 1 shows the risk of oral cancer incidence in the high-risk population who had undergone oral cancer screening. It can be observed that when other variables were controlled, the risk of oral cancer incidence increased as the time elapsed after oral cancer screening increased, with the extent of increase also increasing with time. The accuracy of the model in predicting the risk of oral cancer incidence in the high-risk population was assessed using the AUC (Figure 2). The calculated AUC value for the predictive model was 0.73, which indicates that the model has a good predictive ability [61]. The sensitivity and specificity was 77.1% and 56.4%, respectively. Positive predictive value was 63.9% and negative predictive value was 71.1%, indicating that this model has a good level of validity.

## 4. Discussion

The annual oral cancer incidence rate among the screened individuals was 96 per 100,000 persons—such a high incidence is significantly higher than in other countries [2,3,4,5]. Our results indicate that the annual oral cancer incidence rate per 100,000 persons among individuals with positive screening results was 6.5 times higher than among all individuals screened. This shows that the implementation of the oral cancer screening program is beneficial toward the early diagnosis of oral cancer at the early stage. According to the National Cancer Prevention and Control Program of Taiwan, the follow-up rate for cases of positive oral cancer screening results should reach 80% by 2018 [62]. As of the end of 2018, the rate of confirmed diagnosis of oral cancer among high-risk individuals aged ≥30 years with positive screening results was as high as 79.2%. However, close to 21% of cases with abnormalities did not make return hospital visits for follow-up [63], and an investigation revealed that this was attributed to the lack of close monitoring and subsequent follow-up of the cases by medical institutions.

When the relationships between incidence rate and the various SES-related factors were examined, we found that the incidence rate decreased with increasing educational level; this result is consistent with previous research findings [22,34,35,36,64]. Therefore, incidence rates decrease as income levels increase, which is similar to the results of previous studies [33,35,40,41,45,47,49,50,56,65,66]. Individuals aged 45–64 years had a higher risk of oral cancer incidence than the younger population (HR = 1.88–1.99), but the risk of cancer incidence was not significantly higher in individuals aged ≥65 years than in the younger population. This may be related to the greater number of competing causes of death in the population aged ≥65 years. When the influence of health-related behaviors was investigated, it was found that people who chew betel nuts are 2.12 times more likely to develop oral cancer if they use smoking as a baseline for comparison (HR = 2.12); the results showed that chewing betel nut had a significantly higher oral cancer incidence risk than those who only had a habit of smoking. Individuals who had concurrent habits of smoking and betel nut chewing had a significantly higher oral cancer incidence rate than those who only had a habit of smoking (HR = 2.29). This indicates that betel nut chewing exerts a significantly greater influence on the rate of oral cancer incidence than smoking [18,31,53,55]. Individuals with diabetes (HR = 1.20), other cancers (HR = 1.91), and CCI >2 (HR = 1.27) also had a higher relative risk of oral cancer. Based on the above results, confirmed diagnoses should be performed in a timely manner for individuals who have diabetes or other cancers to achieve diagnosis and appropriate treatment at the early stage and to increase the survival rate of these individuals.

To assess the accuracy of our predictive model for the risk of oral cancer incidence in the high-risk population, a ROC curve was plotted and the AUC calculated. An AUC of 0.73 was obtained, which indicates that the model has good predictive ability for the risk of oral cancer incidence [61]. Studies from India related to the upper gastrointestinal tract showed that the prediction of cancer risk based on the total risk score of chewing tobacco, smoking, drinking, and other unhealthy lifestyle habits in the general population has a high level of predictive power and validity [30,67]. This shows that unhealthy lifestyle habits are indeed an important indicator for predicting the risk of upper gastrointestinal cancer.

To prevent the hazardous effects of oral cancer on the health of the public, the Health Promotion Administration of Taiwan launched an oral cancer screening program in 1999. The program was further expanded in June 2013 to include individuals aged ≥30 years with the habits of betel nut chewing and smoking as well as indigenous people aged 18 years and above with a habit of betel nut chewing with the aim of achieving early detection and administration of appropriate treatment [68]. The World Health Organization (WHO) estimated that the probability of preventing cancer-related death may exceed 30% with the maintenance of regular good lifestyle habits and reduced contact with carcinogens [69]. As oral mucosal examinations can effectively reduce the oral cancer mortality rate by 40% and the five-year survival rate of oral cancer may be as high as 76.7% if detected at stage 0 through oral mucosal examinations, timely oral cancer screening is extremely important. In addition, from the perspective of health promotion, the development of an oral cancer risk model targeted toward the high-risk population with smoking and betel nut chewing habits will be beneficial to the improvement of health-related behaviors and reduction in the incidence and mortality rate due to oral cancer among these individuals.

As oral cancer has multiple etiologies and is closely linked to health-related behaviors, the incidence of oral cancer can only be effectively reduced through health education and preventive healthcare in the public. Notably, individuals who have a habit of betel nut chewing may also have concurrent smoking and drinking habits, which may limit the effects arising from the control of a single risk factor, e.g., efforts to curb smoking may result in excessive drinking instead. Therefore, measures targeting the control of multiple risk factors and the reduction of unhealthy lifestyle habits in the public should be considered when formulating health policies.

This study has some limitations. First, only the data pertaining to subjects who fulfilled the requirements of this study were extracted from the national database on cancer screening. Therefore, we could not retrieve data relevant to the parents of the subjects, such as medical history and cancer-related data. This could have affected the predictive ability of the oral cancer incidence risk prediction model established in this study. Second, as the subjects of the present study were selected based on oral cancer screening records, the model established in this study can only be used to predict oral cancer incidence among individuals who fulfill the criteria for oral cancer screening, i.e., individuals who have a habit of smoking or chewing betel nut. Because the oral cancer screening policy at the time did not list drinking habits as a condition for oral mucosal examinations, the present study did not take this factor into consideration. Therefore, the model is not applicable to individuals without smoking and betel nut chewing habits.

## 5. Conclusions

This result clearly indicates that the screening policy is indeed beneficial to the detection of oral cancer at the early clinical stage. In addition, our results showed that the oral cancer incidence rate was significantly higher among betel nut chewers than among smokers and significantly higher among individuals with concurrent habits of smoking and betel nut chewing than among individuals with either habit. Therefore, effective curbing of betel nut chewing should be the priority target in the prevention and control of oral cancer in Taiwan. As the control of a single risk factor produces limited effects, the adoption of measures aimed at controlling multiple risk factors is recommended when formulating health policies [53,70].

## Figures and Tables

**Figure 1 ijerph-17-00665-f001:**
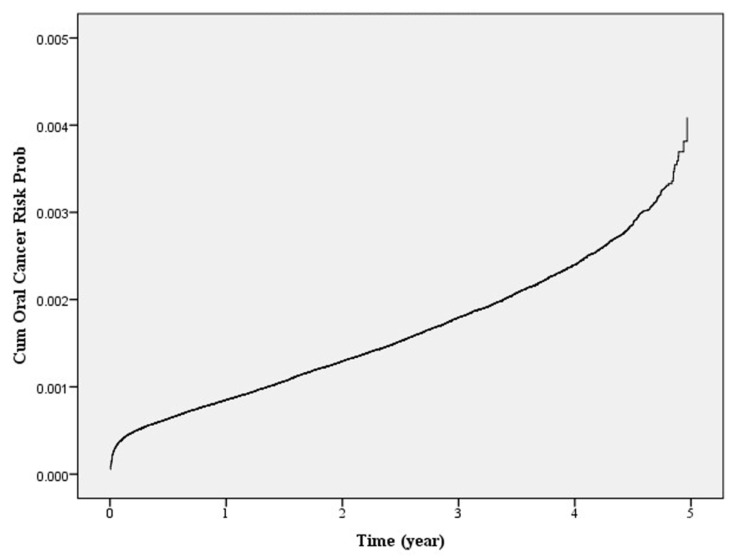
Risk of oral cancer in the high-risk population who had undergone oral cancer screening.

**Figure 2 ijerph-17-00665-f002:**
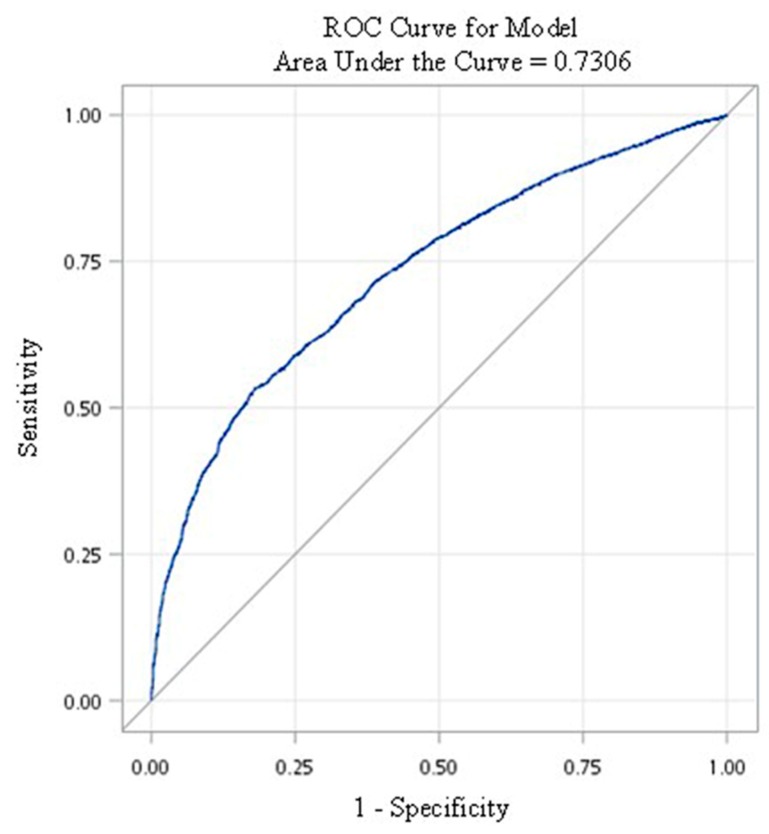
Receiver operating characteristic (ROC) curve of the predictive model for the high-risk population who had undergone oral cancer screening.

**Table 1 ijerph-17-00665-t001:** Basic information and rate of oral cancer incidence in the high-risk population screened for oral cancer.

Variable	All Screened Persons					Positive Cases				
	*N*	*n*	Mean	IDR ^a^	*p*-Value ^b^	*N*	*n*	Mean	IDR ^a^	*p*-Value ^b^
Total	1,719,191	6275	3.80	96	-	117,697	2777	3.78	624	-
Sex										
Female	347,477	267	3.77	20	-	8163	91	3.78	295	-
Male	1,371,714	6008	3.81	115	<0.001	109,534	2686	3.78	649	<0.001
Age, years										
<45	613,275	1306	3.85	55	-	42,531	652	3.84	400	-
45–54	445,594	2085	3.84	122	<0.001	35,937	956	3.80	700	<0.001
55–64	328,320	1757	3.81	140	<0.001	24,712	793	3.75	855	<0.001
≥65	332,002	1127	3.66	93	<0.001	14,517	376	3.62	716	<0.001
Educational level										
Elementary and below	399,726	1915	3.72	129	-	24,082	768	3.68	866	-
Junior high school	448,680	1987	3.82	116	<0.001	35,535	930	3.78	692	<0.001
Senior high school	561,113	1971	3.83	92	<0.001	43,869	912	3.81	545	<0.001
Tertiary	293,723	401	3.83	36	<0.001	13,600	166	3.85	317	<0.001
Unknown	15,949	1	3.93	2	<0.001	611	1	3.93	42	0.002
Marital status										
Married	1,224,584	4614	3.82	99	-	83,139	2030	3.80	642	-
Divorced	176,592	869	3.78	130	<0.001	15,254	393	3.74	688	0.222
Widowed	89,821	251	3.59	78	<0.001	3799	90	3.56	666	0.736
Unmarried	211,957	540	3.79	67	<0.001	14,869	263	3.76	471	<0.001
Unknown	16,237	1	3.93	2	<0.001	636	1	3.92	40	0.006
Indigenous peoples										
No	1,677,465	6126	3.80	96	-	114,451	2707	3.78	625	-
Yes	41,726	149	3.77	95	0.865	3246	70	3.69	585	0.582
Degree of urbanization										
1	318,859	965	3.79	80	-	20,691	443	3.78	566	-
2	424,799	1412	3.79	88	0.024	28,949	652	3.78	595	0.418
3	359,434	1303	3.81	95	<0.001	24,656	580	3.79	621	0.147
4	312,636	1223	3.82	102	<0.001	21,004	511	3.79	641	0.056
5	61,900	293	3.80	124	<0.001	4174	110	3.73	706	0.039
6	129,846	573	3.82	116	<0.001	9090	236	3.73	696	0.011
7	111,717	506	3.79	119	<0.001	9133	245	3.79	708	0.005
Monthly salary										
Low income ^c^	25,565	158	3.69	167	-	2160	68	3.66	860	-
<17,280	325,769	1177	3.80	95	<0.001	23,243	552	3.77	630	0.015
17,281–22,800	638,031	2688	3.77	112	<0.001	45,604	1172	3.74	688	0.073
22,801–28,800	135,455	565	3.82	109	<0.001	9616	247	3.79	678	0.082
28,801–36,300	168,481	608	3.83	94	<0.001	11,630	271	3.81	612	0.012
36,300–45,800	206,153	663	3.85	84	<0.001	14,264	303	3.85	551	0.001
≥45,801	219,737	416	3.85	49	<0.001	11,180	164	3.87	379	<0.001
Health-related behaviors										
Smoking	618,732	1007	3.78	43	-	23,560	316	3.76	356	-
Betel nut chewing	119,804	354	3.75	79	<0.001	5204	162	3.76	827	<0.001
Both	980,655	4914	3.82	131	<0.001	88,933	2299	3.79	683	<0.001
Diabetes										
No	1,533,980	5271	3.81	90	-	101,563	2334	3.79	607	-
Yes	185,211	1004	3.72	146	<0.001	16,134	443	3.73	735	<0.001
Other cancers ^d^										
No	1,672,783	5967	3.82	93	-	114,472	2700	3.80	621	-
Yes	46,408	308	3.35	198	<0.001	3225	77	3.20	746	0.114
CCI										
0	1,370,338	4634	3.83	88	-	90,473	2099	3.81	609	-
1	233,794	1012	3.76	115	<0.001	18,010	425	3.75	629	0.551
≥2	115,059	629	3.53	155	<0.001	9214	253	3.56	770	<0.001
Other catastrophic illnesses ^e^										
No	1,657,132	5997	3.81	95	-	112,708	2660	3.79	623	-
Yes	62,059	278	3.64	123	<0.001	4989	117	3.60	651	0.645

^a^ per 100,000 person years; ^b^ univariate Poisson regression; ^c^ In this study, “low income” was based on the definition of low income households in Taiwan; ^d^ In this study, “other cancer” is defined as other cancers after excluding oral cancer in the Registry for Catastrophic Illness Patients Database; ^e^ In this study, “other catastrophic illnesses” are defined as, in addition to malignant cancer and type 1 diabetes (listed as independent variables), the other 28 categories of diseases registered in the Registry for Catastrophic Illness Patients Database (e.g., dialysis, rare diseases, stroke, hemophilia, etc.).

**Table 2 ijerph-17-00665-t002:** Bivariate analysis of the high-risk populations with and without oral cancer who had undergone oral cancer screening (observed until 2014).

Variable			No cancer		Cancer		
	*n*	%	*n*	%	*n*	%	*p*-Value ^a^
Total	1,719,191	100.00	1,712,916	99.64	6275	0.36	-
Sex							<0.001
Female	347,477	20.21	347,210	99.92	267	0.08	
Male	1,371,714	79.79	1,365,706	99.56	6008	0.44	
Age, years							<0.001
<45	613,275	35.67	611,969	99.79	1306	0.21	
45–54	445,594	25.92	443,509	99.53	2085	0.47	
55–64	328,320	19.10	326,563	99.46	1757	0.54	
≥65	332,002	19.31	330,875	99.66	1127	0.34	
Educational level							<0.001
Elementary and below	399,726	23.25	397,811	99.52	1915	0.48	
Junior high school	448,680	26.10	446,693	99.56	1987	0.44	
Senior high school	561,113	32.64	559,142	99.65	1971	0.35	
Tertiary	293,723	17.08	293,322	99.86	401	0.14	
Unknown	15,949	0.93	15,948	99.99	1	0.01	
Marital status							<0.001
Married	1,224,584	71.23	1,219,970	99.62	4614	0.38	
Divorce	176,592	10.27	175,723	99.51	869	0.49	
Widowed	89,821	5.22	89,570	99.72	251	0.28	
Unmarried	211,957	12.33	211,417	99.75	540	0.25	
Unknown	16,237	0.94	16,236	99.99	1	0.01	
Indigenous peoples							0.763
No	1,677,465	97.57	1,671,339	99.63	6126	0.37	
Yes	41,726	2.43	41,577	99.64	149	0.36	
Degree of urbanization							<0.001
1	318,859	18.55	317,894	99.70	965	0.30	
2	424,799	24.71	423,387	99.67	1412	0.33	
3	359,434	20.91	358,131	99.64	1303	0.36	
4	312,636	18.19	311,413	99.61	1223	0.39	
5	61,900	3.60	61,607	99.53	293	0.47	
6	129,846	7.55	129,273	99.56	573	0.44	
7	111,717	6.50	111,211	99.55	506	0.45	
Diabetes							<0.001
No	1,533,980	89.23	1,528,709	99.66	5271	0.34	
Yes	185,211	10.77	184,207	99.46	1004	0.54	
Other cancers							<0.001
No	1,672,783	97.30	1,666,816	99.64	5967	0.36	
Yes	46,408	2.70	46,100	99.34	308	0.66	
CCI							<0.001
0	1,370,338	79.71	1,365,704	99.66	4634	0.34	
1	233,794	13.60	232,782	99.57	1012	0.43	
≥2	115,059	6.69	114,430	99.45	629	0.55	
Other catastrophic illnesses							<0.001
No	1,657,132	96.39	1,651,135	99.64	5997	0.36	
Yes	62,059	3.61	61,781	99.55	278	0.45	
Health-related behaviors							<0.001
Smoking	618,732	35.99	617,725	99.84	1007	0.16	
Betel nut chewing	119,804	6.97	119,450	99.70	354	0.30	
Both	980,655	57.04	975,741	99.50	4914	0.50	
Monthly salary							<0.001
Low income	25,565	1.49	25,407	99.38	158	0.62	
<17,280	325,769	18.95	324,592	99.64	1177	0.36	
17,281–22,800	638,031	37.11	635,343	99.58	2688	0.42	
22,801–28,800	135,455	7.88	134,890	99.58	565	0.42	
28,801–36,300	168,481	9.80	167,873	99.64	608	0.36	
36,300–45,800	206,153	11.99	205,490	99.68	663	0.32	
≥45,801	219,737	12.78	219,321	99.81	416	0.19	

^a^ log-rank test.

**Table 3 ijerph-17-00665-t003:** Risk of oral cancer incidence and related factors in the high-risk population who had undergone oral cancer screening.

Sex	Adjusted HR	95% CI		*p*-Value ^a^
Female (ref)	-	-	-	-
Male	5.72	5.04	6.49	<0.001
Age, years				
<45 (ref)	-	-	-	-
45–54	1.88	1.75	2.02	<0.001
55–64	1.93	1.77	2.09	<0.001
≥65	1.05	0.95	1.16	0.375
Educational level				
Elementary and below (ref)	-	-	-	-
Junior high school	0.89	0.83	0.96	0.003
Senior high school	0.76	0.71	0.82	<0.001
Tertiary	0.37	0.33	0.42	<0.001
Unknown	0.73	0.00	2597.95	0.939
Marital status				
Married (ref)	-	-	-	-
Divorced	1.35	1.25	1.45	<0.001
Widowed	1.07	0.93	1.22	0.341
Unmarried	0.86	0.79	0.95	0.003
Unknown	0.03	0.00	90.44	0.379
Indigenous peoples				
No (ref)	-	-	-	-
Yes	1.02	0.86	1.21	0.821
Degree of urbanization				
1 (ref)	-	-	-	-
2	1.04	0.96	1.13	0.335
3	0.97	0.89	1.05	0.454
4	0.92	0.85	1.01	0.068
5	0.96	0.84	1.10	0.552
6	0.97	0.87	1.08	0.620
7	1.00	0.89	1.11	0.956
Monthly salary				
Low income (ref)	-	-	-	-
<17,280	0.75	0.63	0.89	0.001
17,281–22,800	0.74	0.63	0.87	<0.001
22,801–28,800	0.74	0.62	0.89	0.001
28,801–36,300	0.65	0.55	0.78	<0.001
36,300–45,800	0.55	0.46	0.66	<0.001
≥45,801	0.44	0.37	0.54	<0.001
Health-related behaviors				
Smoking (ref)	-	-	-	-
Betel nut chewing	2.12	1.87	2.39	<0.001
Both	2.29	2.14	2.46	<0.001
Diabetes				
No (ref)	-	-	-	-
Yes	1.20	1.12	1.29	<0.001
Other cancer				
No (ref)	-	-	-	-
Yes	1.91	1.70	2.14	<0.001
CCI				
0 (ref)	-	-	-	-
1	1.08	1.01	1.16	0.025
≥2	1.27	1.16	1.39	<0.001
Other catastrophic illnesses				
No (ref)	-	-	-	-
Yes	1.06	0.94	1.20	0.352

^a^ Cox PH model.

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
