# Peer review of "Assessment of the Risk of Oral Cancer Incidence in A High-Risk Population and Establishment of A Predictive Model for Oral Cancer Incidence Using A Population-Based Cohort in Taiwan"

_ijerph, 2020, doi:10.3390/ijerph17020665_

Round 1

Reviewer 1 Report

I would like to congratulate the authors for writing a very good manuscript. I have attached my comments to address for the authors. There are several good and relevant references missing from the paper. 

Overall, the paper should be proof edited for english language.

Reviewer 2 Report

This manuscript may be of interest for the readers of International Journal of Environmental Research and Public Health. However, I would like to make two points.

1. This study focused on a high-risk population. Why did the authors include only individuals who had a habit of smoking or betel nut chewing? For instance, it is accepted that alcohol consumption is a risk factor of oral cancer (de Menezes RF, Bergmann A, Thuler LC. Alcohol consumption and risk of cancer: a systematic literature review. Asian Pac J Cancer Prev. 2013;14(9):4965-72). Therefore, the individuals who are heavy drinker may also be a high-risk population.

2. In statistical analyses, the AUC was determined to assess the ability of the model to predict the risk of oral cancer incidence. What model did the authors assess? Clarify this point.

Reviewer 3 Report

Please see my remarks

Round 2

Reviewer 1 Report

The authors have done well with my comments. I approve the amended version.

Author Response

Thank you for your support.

Reviewer 2 Report

The authors addressed my concerns adequately. 

Author Response

Thank you for your support.

Reviewer 3 Report

Please see my remarks

Author Response

Dear Reviewer:

   We have revised the manuscript according to your comments. Thank you.
